# Grounded Decoding: Guiding Text Generation with Grounded Models for Embodied Agents

**Wenlong Huang**[1]*, **Fei Xia**[2], **Dhruv Shah**[3], **Danny Driess**[2], **Andy Zeng**[2],
**Yao Lu**[2], **Pete Florence**[2], **Igor Mordatch**[2], **Sergey Levine**[2,3], **Karol Hausman**[2], **Brian Ichter**[2]
[1]Stanford University, [2]Google Deepmind, [3]UC Berkeley
grounded-decoding.github.io

## Abstract

Recent progress in large language models (LLMs) has demonstrated the ability to learn and leverage Internet-scale knowledge through pre-training with autoregressive models. Unfortunately, applying such models to settings with embodied agents, such as robots, is challenging due to their lack of experience with the physical world, inability to parse non-language observations, and ignorance of rewards or safety constraints that robots may require. On the other hand, language-conditioned robotic policies that learn from interaction data can provide the necessary grounding that allows the agent to be correctly situated in the real world, but such policies are limited by the lack of high-level semantic understanding due to the limited breadth of the interaction data available for training them. Thus, if we want to make use of the semantic knowledge in a language model while still situating it in an embodied setting, we must construct an action sequence that is *both* likely according to the language model and also realizable according to grounded models of the environment. We frame this as a problem similar to probabilistic filtering: decode a sequence that both has high probability under the language model and high probability under a set of grounded model objectives. We demonstrate how such grounded models can be obtained across three simulation and real-world domains, and that the proposed decoding strategy is able to solve complex, long-horizon embodiment tasks in a robotic setting by leveraging the knowledge of both models.

## 1 Introduction

Recent works have demonstrated robots that are increasingly proficient at understanding and acting upon natural language, whether through planning or conditioned policies. Complementing such progress, the field of natural language processing has recently seen large language models (LLMs) become ubiquitously used as pre-trained or few-shot prompted models, due to their impressive few-shot performance and vast knowledge-base. These LLMs have efficiently learned from web-scale data through autoregressively modeling the probability distribution over text tokens and thus generate text. However, the nature of this process is such that applying such models to embodied settings remains a challenge. They have not interacted with their environment, lack observability of non-language observation modalities (e.g., images), and may not know what is safe or possible for a particular embodiment.

Determining how to execute long-horizon behaviors based on high-level verbal commands is one particular area of robotics where the rich semantic knowledge in large language models can be especially useful. This problem combines elements of semantic reasoning and planning: the robot

---

*Work done as an intern at Google.

37th Conference on Neural Information Processing Systems (NeurIPS 2023).

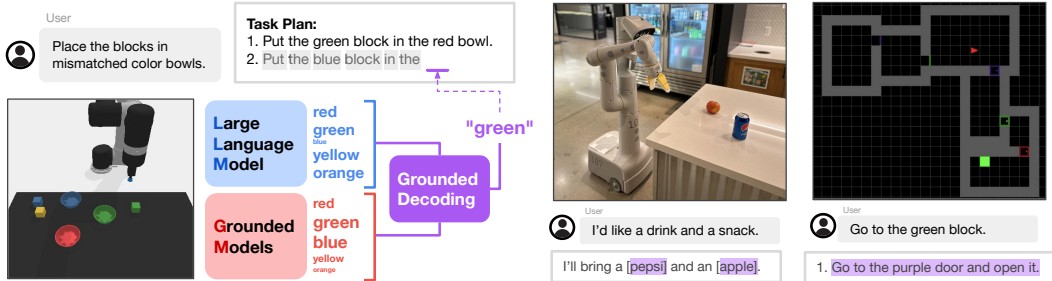

Figure 1: Grounded Decoding solves robotic tasks by taking an instruction as input and selecting tokens that have high probability under a **Large Language Model (LLM)** and a set of **Grounded Models (GM)**. Thus, it leverages the open-vocabulary and semantic knowledge of LLMs while being grounded in the environment and in the robot's capabilities. Furthermore, the whole process does not require expensive fine-tuning of the LLM.

must understand the instruction, determine the steps needed to fulfill it, and also determine how to sequence those steps appropriately given its capabilities and the current state of the environment. However, this is not a problem that can be solved purely with semantics, as it requires sufficient grounding to understand how the task should be performed *in context* – for example, in the example in Figure 1, the language model alone has no way of knowing which block to pick up because this requires knowledge of which blocks are present, and also what manipulations the robot is capable of performing on them. Thus, although a language model can assign probabilities for how likely various steps are to correspond to the desired task *semantically*, the constraints of the planning problem must also enter into the process. These constraints could themselves be represented as probabilities that mirror the token probabilities generated by a language model, reflecting their applicability to the current environment rather than their semantic likelihood. We can frame this as a problem similar to probabilistic filtering: decode a sequence (i.e., a task description) that both has a high probability under the language model and a high probability under a grounded model that predicts how applicable this sequence is to the current scene.

Herein, we present **Grounded Decoding (GD)**, a scalable, general approach to planning with LLMs embodied domains. Grounded Decoding jointly decodes the token probability of an LLM and token probabilities from token-conditioned, robotic functions, such as affordance functions capturing the abilities of a robot given its embodiment, safety functions, or more. By guiding the LLM directly at its output, Grounded Decoding enables a general and flexible family of planning algorithms that combines LLM's strength of *long-horizon* and *semantic* reasoning and grounded models' strength of *local* and *embodiment* grounding.

Our contributions are as followed: 1) we present a robot-centric formulation for decoding language models to perform long-horizon robotic tasks with token-conditioned grounded models, 2) we demonstrate techniques for learning such grounded models, serving different purposes such as affordances and safety requirements, and 3) we show empirical evidence, across three simulation and real-world domains, that the proposed method performs strongly on a wide range of tasks while also significantly outperforming prior methods in efficiency.

## 2   Related Work

**Guided Decoding for Language Models.** Decoding strategies for large language models is an active area of research within natural language processing [77, 87, 25, 85, 38]. A number of recent works have focused on developing decoding heuristics for natural text generation [49, 35, 48, 18, 25, 6, 36, 63]. Another line of works use external classifiers for maximizing certain language-space utilities when decoding language models [71, 92, 26, 39, 37, 21, 38, 4, 12, 23]. Most closely related to our work are classifier-guided decoding methods developed for offline domains, such as image captioning [78, 74] and task-oriented dialog [72, 83]. However, extensions to embodied domains, which we investigate exclusively in this work, remain non-trivial because grounding in embodied domains is bounded by the abilities of the agent and by environment state transition as the agent actively interacts with the environment.

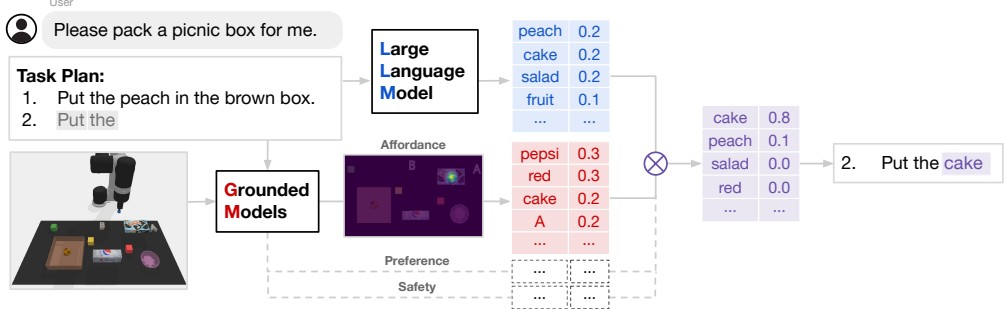

Figure 2: Overview of **Grounded Decoding**. Given a *free-form* language instruction, a **language model** and **grounded models** jointly decide the next candidate token to be decoded by **combining** their respective likelihood. Language model proposes likely tokens that produce goal-directed and coherent long-horizon behaviors, while grounded models connect them to the physical scene, through a flexible composition of multiple objective functions from multiple modalities, such as affordance, preferences, and safety.

**Embodied and Multimodal Language Models.** Training language models to understand embodiment is an active area of research. Training multimodal models can enable some degree of embodied reasoning, such as understanding images and videos [9, 40, 76, 3]. Directly fine-tuning language models to output actions has also been investigated [75, 55, 66]. Lastly, training downstream models on language model embeddings shows promise [46, 52, 24, 94, 60, 41]. In this work, we investigate leveraging large frozen language models for embodied applications [29, 2, 96, 8, 64, 30, 42, 70, 47, 27, 58, 73, 44, 43, 14, 16, 82, 93, 89, 45, 56], with grounded models to provide domain-specific grounding during decoding process.

**Comparison to SayCan.** The most closely related work to our work is SayCan [2]. SayCan uses a large language model and a value function to select robotic skills among a constrained set of primitives. This constrained set of primitives enables SayCan to use the so-called "scoring-mode" of the LLM to get the probability of a skill being useful to a high-level instruction. This requirement to consider only a fixed and enumerated set of primitives limits the applicability of SayCan in scenarios with many possible skills, such as open vocabulary or combinatorial tasks. Grounded Decoding on the other hand jointly decodes the LLM and the grounded model at the token level, allowing for expressive decoding with an open vocabulary. Furthermore, SayCan considers only grounding functions derived from RL-trained value functions for affordance grounding functions, while Grounded Decoding explores many types of grounding functions to propose a broad family of algorithms.

**Task and Motion Planning.** Task and motion planning [33] seeks to solve high-level instructions via sequencing tasks in dynamically feasible manner. Research within this area generally focuses on symbolic planning [19] or optimization-based [79] approaches. Machine learning has increasingly been used to accelerate planning and enable new domains [91, 61, 69, 20, 17, 28, 90, 31, 1, 65, 51, 86, 88, 15]. However, planning constraints are often explicitly specified for TAMP methods. In contrast, we specify constraints as (learned) probabilities, which are baked into the decoding process and provided by domain-specific grounded models.

## 3 Grounded Decoding

### 3.1 LLMs and Grounding Models

**Large Language Models.** LLMs are trained to predict the probability $p(W)$ of a text sequence $W$, represented as a sequence of tokens $W = w_{1:N} = (w_1, \dots, w_N)$. The tokens are elements of a fixed vocabulary $\mathcal{W}$. Typical neural architectures factorize the joint probability into $p(W) = \prod_{n=1}^{N} p_{\text{LLM}}(w_n | w_{1:n-1})$, where $p_{\text{LLM}}$ is predicted by a transformer network [81]. Given $p_{\text{LLM}}$, generating a text consisting of $N$-many tokens, the so-called decoding process, can be seen as the optimization problem $\arg\max_{w_{1:N} \in \mathcal{W}} \prod_{n=1}^{N} p_{\text{LLM}}(w_n | w_{1:n-1})$, which in practice is solved, e.g., using greedy search , beam search, or sampling strategies. To further ensure the LLM is solving a desired task, one typically starts with a given text, the so-called *prefix* or *prompt*, that describes the task, and then the LLM completes this task in its decoding process.

**Grounding Functions.** We use the concept of grounding functions, $p_{\text{G}}(w_{1:n}|s)$, which seek to model a probability of tokens $w_{1:n}$ given (potentially non-textual) state $s \in \mathcal{S}$. This state is intended to capture the embodiment of the robot and the environment, which may be an image, proprioception of the robot, or the environment state. Thus the grounding function models probabilities relevant to the robot embodiment and environment, such as whether the tokens are possible for the robot to execute given the state (affordances), or other values like safety, cost, or user preferences.

### 3.2  Problem formulation.

Given an instruction in language $\ell$, we look at the problem of using an LLM to decode a language plan $w_{1:N}$, which is typically done by finding the most likely tokens under the probability distribution predicted by the LLM, $p_{\text{LLM}}(w_{1:N}|\ell)$, with $\ell$ being the prefix. However, based on the instruction $\ell$ as the prefix alone, the LLM can easily generate text that is not grounded in the physical state of the environment, rendering such plans useless in the real world. In order to *ground* the language model in an actual physical embodiment, we propose *Grounded Decoding* (GD): The main idea of GD is to guide the generation of token sequences with *grounding function(s)* that are conditioned on the embodiment of the system.

Formally, let $s \in \mathcal{S}$ denote a representation of the state of the world. Then, GD attempts to generate text that is consistent with both the instruction $\ell$ *and* the physical state $s$:

$$w_{1:N}^* = \arg \max_{w_{1:N}, w_n \in \mathcal{W}} p_{\text{GD}}(w_{1:N}|s, \ell) \tag{1}$$

To leverage the Internet-scale knowledge of LLMs, we factorize $p_{\text{GD}}(w_{1:N}|s, \ell)$ as follows [2]:

$$p_{\text{GD}}(w_{1:N}|s, \ell) = \frac{p(s, \ell|w_{1:N})\, p(w_{1:N})}{p(s, \ell)} \tag{2}$$

$$= \frac{p(s|w_{1:N})p(\ell|w_{1:N})p(w_{1:N})}{p(s, \ell)} \tag{3}$$

$$= \frac{p(w_{1:N}|\ell)p(\ell)p(w_{1:N}|s)p(s)p(w_{1:N})}{p(s, \ell)p(w_{1:N})p(w_{1:N})} \tag{4}$$

$$\propto \frac{p(w_{1:N}|\ell)}{p(w_{1:N})}p(w_{1:N}|s) \tag{5}$$

$$\propto p(w_{1:N}|\ell)p(w_{1:N}|s). \tag{6}$$

To decode autoregressively with the formulation, we factorize above into token decoding:

$$p_{\text{GD}}(w_{1:N}|s, \ell) \propto \prod_{n=1}^{N} p_{\text{LLM}}(w_n|w_{1:n-1}, \ell)\, p_{\text{G}}(w_{1:n}|s). \tag{7}$$

The first term, $p_{\text{LLM}}(w_n|w_{1:n-1}, \ell)$, can be modeled as the probability of the LLM predicting the token for the given instruction $\ell$ appended previously decoded tokens $w_{1:n-1}$ without the state $s$ as input. The second term, $p_{\text{G}}(w_{1:n}|s)$, is the grounding function that is only conditioned on the state $s$ and judges whether the generated text $w_{1:n}$ is consistent with the physical state. The core idea behind this factorization is that LLMs exhibit long-term planning capabilities, while the grounding function guides the planning of the LLM to be possible in the concrete embodied physical world without needing to be informed or capable of reasoning over the long-horizon instruction.

### 3.3  Grounded Decoding

This work investigates grounded decoding exclusively in the context of task planning for embodied agents. Figure 2 visualizes a single step of the simplest greedy search form of GD, and accompanying pseudo-code can be found in Algorithm 1. Given a high-level language instruction and history of executed actions, GD proceeds through a process similar to probabilistic filtering by selecting tokens iteratively that have high probability under the language model and the grounded model. After

---

[2]We make three assumptions for this derivation: 1) $s$ and $\ell$ are marginally independent (Line 3), 2) $s$ and $\ell$ are conditionally independent given $w_{1:N}$ (Line 5), and 3) $p(w_{1:N})$ is uniform over responses (Line 6).

each token is selected, it is appended to the prefix. The process continues until a token in the terminal set $\mathcal{W}_{\text{term}}$ is selected, which could be a period sign "." indicating the end of a single-step skill (e.g., pick-and-place). Then the command $w_{1:i}$ is sent to a language-conditioned policy $\pi(a|s, w_{1:i})$ that executes the action $a$ conditioned on the environment state $s$. Crucially, this grounding function must accept partial commands to enable grounding during decoding.[3] Additionally, we note that GD, in its essence, provides a grounded scoring function; thus, it can be easily extended to any search methods such as beam search, top-k sampling, etc.

---

**Algorithm 1** Grounded Decoding (GD) w/ Greedy Search

---

1: **Given:** state $s$, instruction $\ell$, terminal set $\mathcal{W}_{\text{term}}$
2: **Initialize:** $w = \{\}, n = 0$
3: **while** $w_n \notin \mathcal{W}_{\text{term}}$ **do**
4: $\quad n = n + 1$
5: $\quad w_n = \arg\max_{w_n \in \mathcal{W}} p_{\text{LLM}}(w_n|w_{1:n-1}, \ell)\, p_{\text{G}}(w_{1:n}|s)$
6: **end while**
7: **Return:** $w$

---

### 3.4 Techniques for Obtaining Grounded Models

Unlike language tasks, where a single model is capable of performing general semantic reasoning, a singular grounded model remains an open problem. Indeed, each domain may impose varied environmental and embodiment constraints. Despite these challenges, we present several techniques for obtaining grounded models that can be leveraged in GD's formulation, and validate them in three domains in Section 4.

**Token-Conditioned Value Functions.** Assuming a robot acts with action $a$ according to policy $\pi(a|s, w_{1:n})$, that aims to maximize certain a utility and that the utility captures a task objective, a natural choice that can provide "grounding score" is the action-value function $Q(s, a|w_{1:n})$ as it necessarily captures the embodiment of the robot. Additional objectives, such as task constraints, can also be encapsulated in $Q(s, a|w_{1:n})$ to ensure grounding. Note that unlike the formulation proposed in [2], $w_{1:n}$ cannot be restricted to a fixed repertoire of token sequences. In practice, to obtain a $Q(s, a|w_{1:n})$ that satisfies the requirements, one can train multi-task language-conditioned agents, either through reinforcement learning (Section 4.2) or supervised learning (Section 4.1).

**Multimodal Foundation Models.** A general choice to ground LLMs is through using multimodal foundation models, such as CLIP [57] or open-vocabulary object detectors [22, 34, 50]. Although these models can connect language to other grounded modalities (e.g., vision), they often lack the capability for complex or long-horizon reasoning, and they do not consider embodiment constraints. As a result, to leverage them in the decoding process, they need to constrained to where they are the most applicable rather than always decoding jointly. To this end, we use a prompt-based technique that allows LLMs to choose when to jointly decode (Section 4.3), which we find to be effective in most cases.[4].

**Rule-based Methods.** Another source of grounding may come from features $x = \phi(w_{1:n})$ designed with expert domain knowledge, which can then be used to map $w_{1:n}$ to a "grounding score" using pamametric or non-parametric functions $f(x)$. Such techniques may be most applicable when interpretability and enforcing hard constraints are required, such as safety-critical settings, or when data are scarce, such as cases involving preferences of individual users (as shown in Section 4.1).

### 3.5 Comparisons to Prompt-based Methods

One alternative approach for grounding is including scene information as part of the prompt (e.g., object detection results [96]), which complements the grounding method proposed in this work.

---

[3]As an example, an affordance ground function for a skill "pick up *object*", should emit a high probability for "pick" and "pick up" if any object is able to be picked and collapse to only feasible objects only once the *object* token is decoded.

[4]Emerging multimodal language models [16, 53] provide strong baselines, but they similarly cannot serve general-purpose grounding functions because they are not conditioned on embodiment, except for cases where embodiment data from each individual domain can be used to finetune the LLM [16].

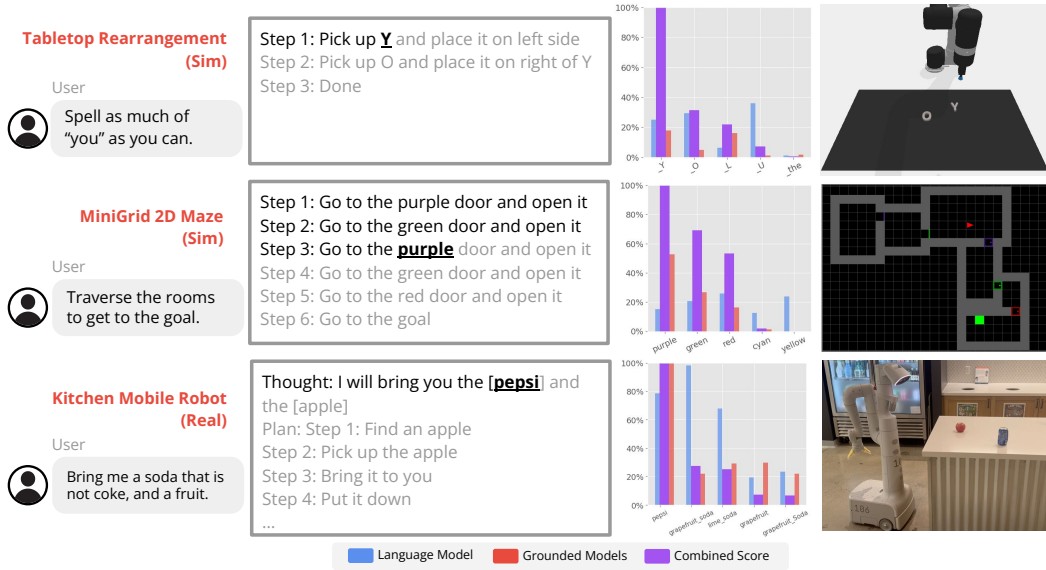

Figure 3: Example rollouts and likelihood of representative tokens under Grounded Decoding objective in three distinct domains: simulated tabletop rearrangement (**top**), Minigrid 2D Maze (**middle**), and real-world kitchen mobile manipulation (**bottom**). Each domain uses different prompts, grounded models, and low-level primitives. The GD formulation is shared across the domains, decoding a pre-trained langauge model with respect to domain-specific grounded models to decompose a *open-ended* instruction into actionable steps.

However, we note that prompting is often insufficient for grounding, as information about the scene and about the capabilities of the robot may not always be succinctly described in the prompt. Such examples include 1) in a block stacking task, a block that has been stacked on cannot be picked, 2) in a navigation task, to open a door, one must have a key and that door must be reachable, and 3) in a mobile manipulation domain, an object may be visible but is out of reach of the manipulator. Therefore, Grounded Decoding is a more general and flexible grounding method that injects *continuous probabilities* during decoding, which may even come from grounding functions from *other modalities* (e.g., vision).

## 4 Experiments

### 4.1 Long-Horizon Tabletop Manipulation

Herein we experiment with a simulated tabletop manipulation environment based on RAVENS [95]. We create a custom set of 20 tasks, with 10 seen tasks and 10 unseen tasks. Seen tasks are used for training (for supervised baseline) or for few-shot prompting. They are grouped by following categories. Detailed breakdown can be found in Appendix A.2.

i. **Letters**: Rearranging alphabetical letters ("sort the letters in alphabetical order").

ii. **Blocks & Bowls**: Rearranging or combining blocks and bowls ("put blocks in matching bowls").

iii. **Box Packing**: Sorting food items and utensils into boxes in accordance with safety constraints and user preferences ("Can you pack the picnic box for me?").

Given only high-level language instructions and top-down visual observation of the environment, Grounded Decoding decodes a sequence of text tokens representing the step command to be executed. Note that because GD generated grounded *free-form* actions, it does not require each step to strictly map to a repertoire of skill as in [29, 2]. After a complete command is generated, it is executed via a pre-trained multi-task language-conditioned CLIPort [67]. An example rollout is shown in Figure 4.To demonstrate the techniques proposed in Section 3.4 to obtain grounding functions, we study the composition of following grounding functions (overall grounding score is calculated as $p_{\mathrm{G}} = \prod_{i=1}^{n} p_i$) depending on the task categories. Refer to the Appendix A.2 for details.

| | CLIPort | | +LLM | +GD (Ours) | |
|---|---|---|---|---|---|
| | **Short** | **Long** | **Ungrounded** | **Greedy** | **Beam** |
| **Seen Tasks** | | | | | |
| **Letters** | 7% | 40% | 20% | 43% | **57%** |
| **Blocks & Bowls** | 2% | 62% | 35% | 60% | **77%** |
| **Box Packing*** | 15% | 28% | 11% | **79** % | 78% |
| **Unseen Tasks** | | | | | |
| **Letters** | 6% | 10% | 19% | 37% | **41%** |
| **Blocks & Bowls** | 6% | 10% | 28% | 44% | **50%** |

Table 1: Tabletop domain success rates.

| | +Skills | | +LLM | +GD (Ours) | |
|---|---|---|---|---|---|
| | **PPO** | **HRL** | **Ungrounded** | **Greedy** | **Beam** |
| **Easy** | 28% | 68% | 96% | **100%** | **100%** |
| **Medium** | 13% | 48% | 87% | 93% | **97%** |
| **Hard** | 6% | 31% | 54% | 78% | **88%** |

Table 2: 2D Maze success rates.

**Affordance Grounding Function (AF).** As the primitive policy CLIPort [67] already acts as an action-value function over the pixel space, we directly leverage its predictions for affordance grounding. In particular, given scene image $s$ and partially-decoded instruction $w_{1:n}$, CLIPort predicts unnormalized logits over the pixel space $\mathbf{u}_{\text{pick}}, \mathbf{u}_{\text{place}} \in \mathbb{R}^{480 \times 640}$, respectively for the pick location and the place location. Therefore, for any given $s$ and $w_{1:n}$, we can calculate the affordance as $p_{\text{AF}}(w_{1:n}|s) = \max_{(x,y) \in 480 \times 640} (\mathbf{u}_{\text{pick}}(x, y) + \mathbf{u}_{\text{place}}(x, y))$.

**Safety Grounding Function (S).** To strictly enforce hard constraints such as safety requirements, we adopt the rule-based method proposed in Section 3.4. In particular, the features $x$ are indicator functions denoting whether knives and red boxes which we use as hazardous objects are involved in an action, i.e., $x = \mathbb{I}[\text{"red" or "knife" in } w_{1:n}]$. We then use constant mappings to convert the features $p_{\text{S}}(w_{1:n}|s) = \frac{\epsilon}{Z}x + \frac{1}{Z}(1 - x)$ to scores of 0 or 1, where $Z$ is the normalizing term and $\epsilon \approx 0$ is a small value for ensuring the joint probability does not collapse to 0.

**Preference Grounding Function (P).** We similarly use rule-based methods for preference grounding, as data of individual users may be scarce to learn a separate model. In particular, we choose two random objects $(o_1, o_2)$ as the preferred objects, i.e., $x = \mathbb{I}[o_1 \text{ or } o_2 \text{ in } w_{1:n}]$. Note that unlike safety functions, preferences often come in the form of "soft requirement". Therefore, the preference grounding function is implemented as $p_{\text{P}}(w_{1:n}|s) = \frac{\alpha}{Z}x + \frac{\beta}{Z}(1-x)$, where we choose $\alpha = 0.5$ and $\beta = 0.1$ for our experiments.

**Baselines.** We study two variants of GD using beam search and greedy search. We also compare to "No Grounding" baseline that decodes only according to language model likelihood. Furthermore, we compare to solitary method CLIPort [67] that directly take in the high-level language instructions without a planner. We consider two variants of CLIPort: 1) "Short" that is trained with only single-step pick-and-place commands, and 2) "Long" that is trained on high-level instructions from the 10 training tasks. For more details, please refer to Appendix A.2.

**Analysis.** Results grouped by each task category are shown in Table 1[5]. Please refer to the Appendix for detailed breakdown. Each method is evaluated on 20 episodes for each task within each task category. Supervised methods, such as CLIPort, are found to perform poorly on unseen tasks. Methods that leverage language model planner show better generalization to unseen tasks but fall short due to lack of grounding. Grounded Decoding achieves the best results by enabling the LLM to plan actions using grounded information and is further improved with beam search.

## 4.2   2D Maze

We further evaluate the long-horizon reasoning of Grounded Decoding for 2D maze-solving on Minigrid [10]. The agent receives a top-down view of the environment along with a natural language instruction. More details can be found in Appendix A.3. The tasks are grouped in three categories:

i. **Easy**: Simple tasks where the horizon is short (10-30 steps) and fully described by the textual instruction, e.g. `OpenDoors` and `PutNext`.

ii. **Medium**: Short and long-horizon tasks (up to 80 steps) with step-by-step textual instructions, e.g. `LockedRoom`.

iii. **Hard**: Complex, long-horizon instructions (over 100 steps) with ambiguous instructions that necessitate multi-step reasoning and efficient exploration, e.g. `MultiRoom` and `BlockedUnlock`.

---

[5]Box Packing tasks are seen during training, but safety and preference requirements are only enforced during evaluation.

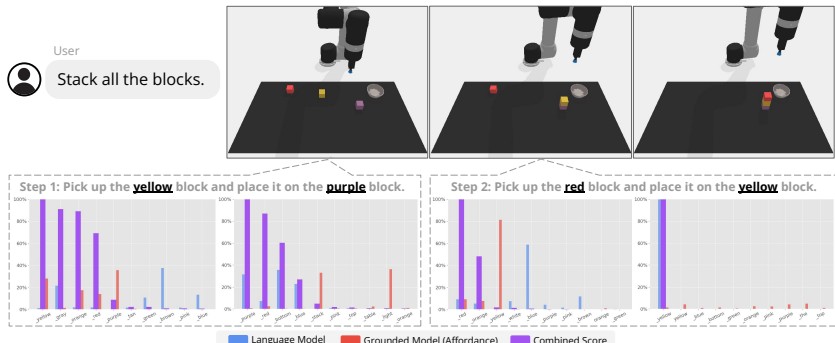

Figure 4: Greedy decoding rollout with GD, where key decoded tokens are shown (*yellow*, *purple*, *red*, *yellow*). Combined scores are normalized to the maximum for visual clarity; others are normalized to their sum.

**Affordance Grounding Function.** Following the recipe from Section 3.3, we train token-conditioned affordance function to be used in GD. The difference is that the grounding function here is the value function from the goal-conditioned policy that is trained with PPO [62] instead of from demonstrations as in CLIPort [67]. The policy performs short-horizon skills such as "Go to red key" or "Open the door" and are conditioned on CLIP embeddings of the skill and an image of the scene. Accordingly, the goal-conditioned value function evaluates the feasibility given the current observation and the (partially) decoded skill.

**Baselines.** We compare the two variants of GD – with greedy and beam search – with 1) a solitary PPO policy [62], 2) a hierarchical RL algorithm which plans over the low-level skills, and 3) a hierarchical method that uses an ungrounded language model for planning [29].

**Analysis.** Table 2 reports the success rate, averaged across 100 episodes of randomly initialized environments. The "flat" RL agent performs poorly in all but the simplest environments, owing to difficulties with understanding the high-level instruction and reasoning over long horizons (often over 100 steps). Planning over low-level skills using hierarchical RL [5] improves this performance, since the high-level decision-making problem is greatly simplified. However, the high-level RL agent still needs to reason over low-level (textual) skills by understanding their underlying capabilities and stitching them together. Using the planning capabilities of large language models to reason over textual skills significantly boosts this performance [29], since the language model can inherit the strong reasoning capabilities from its training data. This tends to be insufficient in challenging environments, however, since the number of *potentially viable* skills may be very large and the LLM has no information about the robot's observations. GD can leverage the learned affordance function (in this case, the goal-conditioned value function) to inform the language model's plans, enabling successful long-horizon reasoning. We further find that beam search improves performance modestly, particularly in long-horizon tasks.

### 4.3 Mobile Manipulation in a Physical Kitchen

Our last environment is a kitchen robot in the real world, and we follow the same implementations of the mobile manipulation platform and skills in SayCan [2]. We perform instruction following tasks, as in [2]. An example task is "Bring an apple", for which the robot needs to plan and execute a sequence of "1. Find an apple, 2. Pick up the apple, 3. Bring it to you. 4. Put it down, 5. Done". We split the tasks into two categories. *Unambiguous* means the instruction explicitly contains the object of interest, and *Ambiguous* means the instruction does not contain the object name. For example, when human asks "bring me the fruit", the robot needs to first determine available fruits. We assume all necessary objects are in the field of view. More details can be found in Appendix A.4.

**Grounded Decoding with Chain-of-thought.** We demonstrate using multimodal foundation models for Grounded Decoding, as proposed in Section 3.4. In particular, we use open-vocabulary object detector owl-vit [50]. Note that because these off-the-shelf models are not trained on robot domain data, we find that it works best by constraining their influence on decoding. We achieve this by making a slight modification to the SayCan algorithm [2]: before generating action plans, we prompt the LLM to generate *visually-grounded* chain-of-thought [84] by giving LLM the option of when to enable grounded decoding and disable grounded decoding, as visualized in Fig. 5. Specifically,

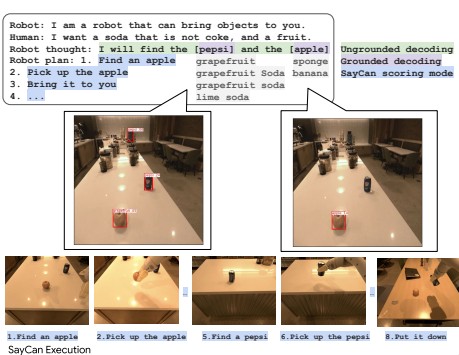

```
Robot: I am a robot that can bring objects to you.
Human: I want a soda that is not coke, and a fruit.
Robot thought: I will find the [pepsi] and the [apple]    Ungrounded decoding
Robot plan: 1. Find an apple    grapefruit         sponge   Grounded decoding
2. Pick up the apple            grapefruit Soda banana       SayCan scoring mode
3. Bring it to you              grapefruit soda
4. ...                          lime soda
```

1.Find an apple   2.Pick up the apple   5.Find a pepsi   6.Pick up the pepsi   8.Put it down

SayCan Execution

Figure 5: Example prompt and rollout in real-world kitchen environment.

| Tasks | GD | | SayCan | |
|---|---|---|---|---|
| | Planning | Execution | Planning | Execution |
| **Unambiguous** | 85% | 57% | 85% | 57% |
| **Ambiguous** | 58% | 44% | 33% | 25% |

Table 3: Success rates in kitchen environment.

| | GD (Greedy) | GD (Beam) | SayCan |
|---|---|---|---|
| **Success Rate** | 50% | 60% | **64%** |
| **Token Count** | **1x** | 4.3x | 113.7x |

Table 4: By avoiding full enumeration of skills, GD is more efficient than SayCan while staying performant.

LLMs can be prompted to generate a left bracket to start decoding jointly with grounded models and a right bracket to revert to ungrounded decoding. After chain-of-thought, we use SayCan scoring mode for decoding the action plans.

**Analysis.** Table 3 shows that GD recovers similar performance on *Unambiguous* tasks, and gain 25% in planning performance on *Ambiguous* tasks. This shows that GD with multimodal foundation models can effectively use *visually-grounded* chain-of-thought to disambiguate abstract tasks.

## 5 Analysis

### 5.1 Comparison to SayCan

In this section, we directly compare GD to SayCan [2], which is related to our method in that both combine language model knowledge and grounded model knowledge (discussed in more detail in Section 2). However, SayCan uses the language model to score all pre-specified options, rendering it inefficient at dealing with large or combinatorial action spaces. In contrast, GD computation considers all possible language token in the autoregressive decoding process, which is *independent* of the size of the action space. Results shown in Table 4 demonstrate that GD is two orders of magnitude more efficient on our tasks, with comparable performance. Furthermore, by decoding at the most basic functioning unit of language, GD's formulation allows open-vocabulary grounding beyond just affordances, e.g. safety, preferences, and multimodal embeddings such as CLIP.

### 5.2 Breakdown of Failure Reasons

Because all hierarchical approaches share an imperfect low-level policy for step execution, the results reported in Table 1 are compounded with both planning failures and low-level policy failure. In Figure 6, we provide failure breakdown analysis for Grounded Decoding and associated baselines. Note that the CLIPort baselines are solitary methods that do not use a planner, so the failures are solely composed of policy failures. As shown in Figure 6, while all planning-based methods use the same underlying low-level policy, Grounded Decoding significantly reduces planning failure by being able to incorporate grounded scene information into the decoding process. Moreover, we observe that despite the shared affordance function across beam and greedy search, the beam search variant performs stronger by being aware of the full-length single-step instructions during decoding.

### 5.3 Grounded Action Manifold

A central goal of this work is to investigate the integration of grounded information into language model decoding to output instructions actionable by a policy. To investigate this, we use a t-SNE [80] plot to illustrate the extent to which grounded models help narrow down the search space for language models. Specifically, we first enumerate all meaningful instructions in the tabletop domain, such as "pick x and place it on y," which are represented as dots in the figure. We then compute the affordance values with respect to four different scenes, where each color represents one scene. Finally, we group the dots using t-SNE and BERT embeddings [13]. Figure 7 shows that grounded

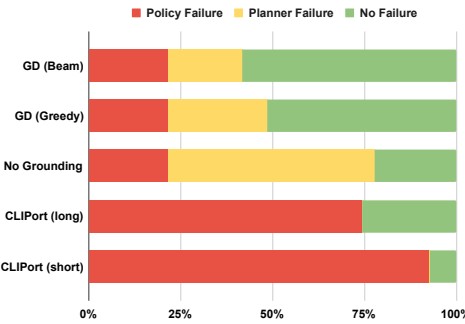

Figure 6: Failure breakdown on tabletop domain. GD achieves lowest planning failure among planning-based methods, among which beam search variant performs the best.

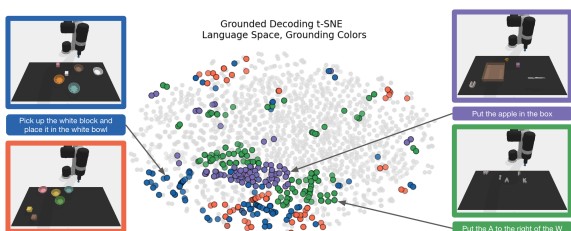

Figure 7: Visualization of actions colored by affordance values in different scenes. Every dot represents a possible action in the tabletop domain, where the majority of the actions are infeasible. We show how grounded models can identify the feasible actions for specific scenes. Notably, these actions are not always clustered in language space, requiring the grounding function to determine what action to perform.

models can effectively identify achievable skills to produce an actionable manifold within the language space and that this grounding is required, as language alone does not perfectly group actionable skills. It is worth noting that while we provide manual enumeration of all possible skills for practical analysis, the full language space is much larger. This highlights the even more pronounced narrowing of the search in the language space.

## 6 Conclusions, Limitations, & Future Works

We presented Grounded Decoding (GD), an approach for leveraging the knowledge and capabilities of large language models in embodied settings through grounding functions, which model the probabilities of tokens given an embodiment. GD resembles probabilistic filtering, by decoding tokens that have high probabilities under the language model *and* under grounded model(s). By guiding the LLM's decoding directly at its output, GD is a general, flexible, and expressive approach to embodied tasks. This is demonstrated on three embodied domains, showing GD is capable of solving complex, long-horizon tasks.

Though quite general and flexible, GD has a few limitations. First, although we present several techniques for obtaining grounding functions in different domains, it remains a question whether a capable and general grounding function can be obtained. We hope that recent progress in large-scale robotics models (e.g. [7] and [59]) can remove this bottleneck, and note that the flexibility of GD allows such progress to be straightforwardly leveraged. Second, prompt engineering is often required to steer LLMs to the desired action space (e.g., likely action verbs, likely present objects). Finally, while not requiring additional training, the joint decoding may be limiting compared to a single model capable of both grounding and language reasoning [3, 9, 16].

This work presented a family of algorithms for grounding LLMs in embodiment, for which there are many avenues for future work. The flexibility of the approach enables many other grounding functions and ways to integrate grounding. Furthermore, the development and integration of a foundation model for grounding would improve performance significantly. Finally, though GD's probabilistic filtering-based approach is quite general, fusing grounding information to the language model *after* each token decoding may be limiting and future works can investigate how such grounding can be elegantly integrated *during* decoding.

### Acknowledgments

The authors would like to acknowledge Pierre Sermanet, Carolina Parada, Jie Tan, Yevgen Chebotar, Vincent Vanhoucke, and Dorsa Sadigh for their feedback and contributions. This work is supported in part by OpenAI academic access program, granted to Wenlong Huang.

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

# A  Appendix

## A.1  Grounded Decoding Implementation Details

We study three different implementations of Grounded Decoding for each of the experimental domains. While each instantiation applied Grounded Decoding to long-horizon planning and behavior synthesis, different components including language models and grounded models are used in each domain, as seen in Table 5. Grounded models used in these domains include Affordance Functions (AF), Safety Functions (S), Preference Functions (P), and Open-Vocabulary Object Detectors (D).

|  | Tabletop Rearrangement (Sim) | MiniGrid 2D Maze (Sim) | Kitchen Mobile Manipulation (Real) |
|---|---|---|---|
| **LLM** | InstructGPT [54] | InstructGPT | InstructGPT + PaLM[11] |
| **Primitives** | CLIPort [67] | PPO [62] | RT-1 [7] |
| **Grounded Models** | AF + S + P | AF | D |

Table 5: Comparison between different versions of GD implemented in three different environments.

## A.2  Implementation Details of Simulated Tabletop Rearrangement

### A.2.1  Tasks

There are a total of 20 tasks (templates of language instructions), listed in Table 6, grouped into three task category: Letters, Blocks&Bowls, and Box Packing. Three categories share a total of 57 objects. For Letters category, the goals are to rearrange the alphabetical letter objects such that they satisfy certain orders specified by the language instructions. At the beginning of each episode, task-relevant objects and a set of 1 to 3 randomly-sampled distractor objects (except for the Letters category) are initialized at random positions on the tabletop with fixed orientations. A minimum 15cm distance is enforced between any two objects to avoid collision and penetration at initialization. To allow for automatic evaluations, a binary reward function is defined for each task using ground-truth state of the objects. Furthermore, we implement scripted policies for each task to collect demonstrations for training the CLIPort baseline. For certain tasks, we also randomize the attributes mentioned in the given instructions, which can be found below:

i. ⟨word⟩: hi, world, left, right, top, down, love, you

ii. ⟨corner/side⟩: left side, top left corner, top side, top right corner, bottom right corner, bottom side, bottom left corner

### A.2.2  Low-level Primitives

We use CLIPort [67] as the low-level primitive that can be invoked by the LLM planner, as it shows promising results of generalization across free-form language instructions. Additionally, since the policy predicts per-pixel affordance, it can be repurposed to serve as grounded models for planning for long-horizon tasks, which we leverage in this work. The single primitive policy is trained on 50,000 pre-collected demonstrations, across 10 training tasks, where each demonstration contains 1) language instruction of the format "pick up [x] and place it on [y]", 2) top-down RGB-D observation of the current scene, 3) expert pick location expressed as pixel coordinates, and 4) expert place location expressed as pixel coordinates. The expert actions are obtained by accessing ground-truth object pose in the simulator. We further apply substring augmentation during training as we find it helps with performance on partial commands (see Section

### A.2.3  Language Model

We use InstructGPT [54] (`text-davinci-002`), accessed through OpenAI API.

### A.2.4  CLIPort Baseline

As CLIPort [67] already takes as input a natural language instruction and is capable of directly outputting robot actions, it bears the question whether we need a high-level planner for completing

long-horizon tasks. To this end, we additionally train two variants of multi-task CLIPort policy on 10 of the total 20 tasks as baselines (see Table 6 for the train/test split). One variant, which we referred as "CLIPort (Short)", is trained only on single-step pick-and-place instructions of the format "pick up [x] and place it on [y]" on the 10 training tasks. The decomposed pick-and-place instructions are obtained from scripted planners. At evaluation time, the policy is fed in only the high-level instructions without any planners. The other variant, which we referred as "CLIPort (Long)", is trained on the high-level instructions from the 10 training tasks (without decomposition from scripted planners). Similarly, at evaluation time, it is fed in only the high-level instructions and evaluated on both seen and unseen instructions. Both variants are trained on 50,000 demonstrations, similar to the Grounded Decoding primitive. The goal of these baselines is to evaluate whether solitary language-conditioned policies can perform well on long-horizon tasks and generalize to new task instructions. Note that the CLIPort baselines are different from the primitive used in Grounded Decoding, although they share the same architecture.

### A.2.5 Full Experimental Results in Simulated Tabletop Domain

Below we show the full list of tasks and the full experimental results in the simulated tabletop domain. Each entry is the average success rate across 20 rollouts. The tasks with blue-colored background are seen tasks and the tasks with orange-colored background are the unseen tasks. Seen tasks may be used for training for supervised baselines (CLIPort) or may be used in the prompt for methods using language model planner. Note that for the "Box Packing" task category, although all tasks were seen in training or the prompts, we enforce additional safety and preference constraints for evaluation only at test time.

| Tasks | $p_G$ | CLIPort | | +LLM | +Grounded Decoding | |
| | | Short | Long | Ungrounded | Greedy | Beam |
|---|---|---|---|---|---|---|
| **Letters** | | | | | | |
| Put the letters in alphabetical order from left to right | AF | 5% | 20% | 10% | 20% | **40%** |
| Spell as much of *word* as you can | AF | 10% | 60% | 30% | 60% | **65%** |
| Separate the vowels from the remaining letters to the bottom side | AF | 5% | 40% | 20% | 50% | **65%** |
| Put the letters in reverse alphabetical order from left to right | AF | 15% | 10% | 15% | **25%** | **25%** |
| Correctly spell out a sport using the present letters | AF | 10% | 10% | 5% | **30%** | **30%** |
| Sort the geometrically symmetrical letters to the bottom side | AF | 5% | 10% | 15% | 35% | **50%** |
| Separate the consonants from the remaining letters to the bottom side | AF | 0% | 0% | **25%** | **25%** | **25%** |
| Sort the letters less than "D" according to ASCII to the bottom side | AF | 0% | 20% | 35% | 70% | **75%** |
| **Blocks & Bowls** | | | | | | |
| Stack all the blocks | AF | 5% | **90%** | 30% | 75% | **90%** |
| Put all the blocks on the *corner/side* | AF | 0% | 65% | 50% | 45% | **70%** |
| Put all the blocks in the bowls with matching colors | AF | 0% | 30% | 25% | 60% | **70%** |
| Put the blocks in the bowls with mismatched colors | AF | 25% | 30% | 45% | 30% | **55%** |
| Put all the blocks in different corners | AF | 0% | 5% | 40% | 50% | **60%** |
| Stack only the blocks of cool colors | AF | 5% | 5% | 20% | **70%** | **70%** |
| Stack only the blocks of warm colors | AF | 0% | 10% | 15% | **45%** | 35% |
| Sort the primary color blocks to the left side | AF | 0% | 0% | 20% | 25% | **30%** |
| **Box Packing*** | | | | | | |
| Pack the objects into the brown box | AF + S | 20% | 40% | 5% | **100%** | 90% |
| Pack the objects into the boxes | AF + S | 10% | 20% | 5% | **75%** | 70% |
| I'd like some snacks on the right side | AF + P | 15% | 20% | 15% | 40% | **55%** |
| Pack me a picnic box | AF + S + P | 15% | 30% | 20% | **100%** | 95% |

Table 6: Full Experimental Results in Simulated Tabletop Rearrangement Tasks. The tasks with blue-colored background are seen tasks and the tasks with orange-colored background are the unseen tasks. *Box Packing tasks are all seen during training, but safety and preference requirements are only enforced during evaluation.

### A.3  Implementation Details of Minigrid 2D Maze

#### A.3.1  Environment Setup

We use the open-source `gym-minigrid` suite of environments to evaluate our method with one simple change — instead of the default observation space which is a $7 \times 7$ egocentric window, our agent has access to *entire grid* — that allows us to simplify the tasks by removing partial observability [65].

#### A.3.2  Tasks

The tasks are grouped in three categories (please see Table 7 for example instructions):

1. **Easy**: Simple tasks where the horizon is short (10-30 steps) and fully described by the textual instruction, e.g. `OpenDoors` and `PutNext`. The short horizon makes them relatively easy for a wide range of HRL algorithms. The instructions for these tasks generally spell out each individual skill, making them particularly easy for high-level planners based on language modeling.

2. **Medium**: Combination of short and long horizon tasks (up to 80 steps) with step-by-step textual instructions, e.g. `LockedRoom`. While being significantly longer, these tasks also tend to have instructions that spell out the low-level tasks (see Table 7).

3. **Hard**: Complex, long horizon instructions (over 100 steps) with short, ambiguous instructions that necessitate multi-step reasoning and efficient exploration, e.g. `MultiRoom` and `BlockedUnlock`. In addition to being long-horizon, the instructions in this case tend to be ambiguous and under-specified, e.g. "traverse through the rooms to get to the goal", which does not provide enough context for any *blind* planning agent.

| Difficulty | Task Name | Example Instruction |
|---|---|---|
| Easy | OpenDoors
PutNext | open door blue, then open door red
move the red ball next to the green box |
| Medium | LockedRoom | get the red key from the purple room, open the red door and go to the goal |
| Hard | MultiRoom
BlockedUnlock | traverse the rooms to get to the goal
pick up the blue box |

Table 7: Example Instructions in Minigrid

#### A.3.3  Language Model

We use InstructGPT [54] (`text-davinci-002`), accessed through OpenAI API. The prompts used can be found in Section A.5.

We found the prompts to be generally sufficient for solving the "seen" tasks, as well as "unseen" tasks, i.e. tasks that do not have an example in the context. Empirically, we did not find any improvements by including more then 3 example tasks in the prompt — we hypothesize that this is likely due to the shared low-level primitives across tasks. For all Minigrid experiments presented in this paper, we used the prompt shown in Section A.5.

#### A.3.4  Low-level Primitives

To train low-level primitives, we train an RL agent to solve a wide range of short-horizon sub-tasks (under 10 steps) that are shared across the various Minigrid tasks — `go to <obj>`, `pick up <obj>`, `drop <obj>`, `open <obj>`. Rather than training individual skills for each of them [65], we train a single multi-task policy that is conditioned on the CLIP embeddings [57] of the task strings. This scheme allows some robustness to synonyms and ambiguous task specifications, and has been widely used in learning language-grounded policies [32, 68].

We train these primitives using PPO [62], as recommended by the environment developers [10]. Each of these skills are trained with a sparse outcome reward (+1 if a trajectory is successful, 0

otherwise). In addition to these low-level skills, we perform a form of hindsight relabeling where "substrings" of the task strings are masked to allow generalization to partial strings, e.g. "go to red" may be interpreted as "go to red key" or "go to red door", and our masking strategy allows the multi-task policy to execute tasks specified by partially complete strings, if necessary.

### A.3.5 Additional Qualitative Results

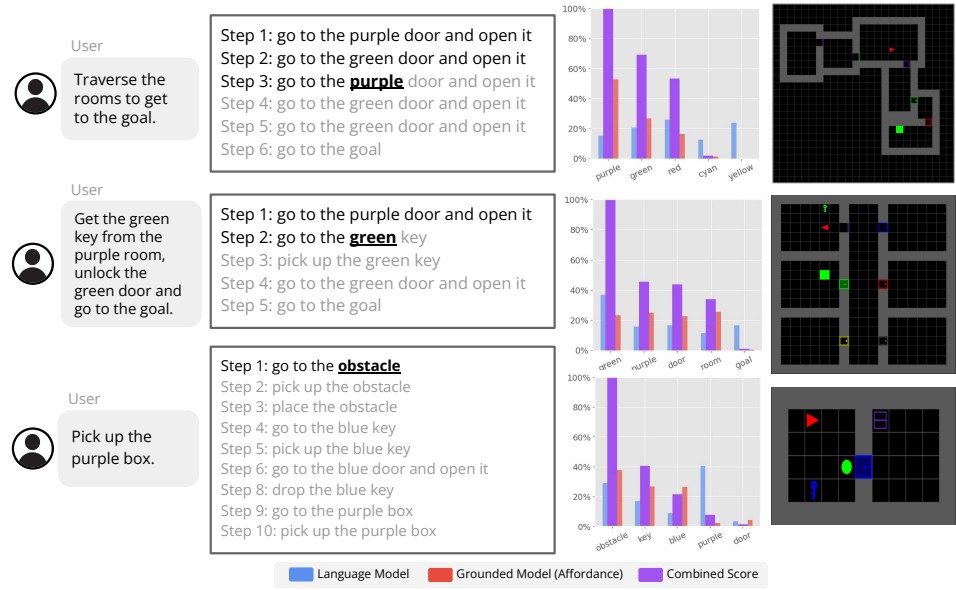

Figure 8: Minigrid Domain

### A.4 Implementation Details of Real-World Mobile Manipulation

#### A.4.1 Tasks

| Instruction |
| --- |
| put an energy bar and water bottle on the table |
| bring me a lime soda and a bag of chips |
| Can you throw away the apple and bring me a coke |
| bring me a 7up can and a tea |
| move an multigrain chips to the table and an apple to the far counter |
| move the lime soda, the sponge, and the water bottle to the table |
| bring me an apple, a coke, and water bottle |

Table 8: **List of unambiguous SayCan instructions**

| Instruction |
| --- |
| I want to wipe some spill. |
| Bring me a fruit |
| Bring me a snack |
| Bring me a bag of chips |
| Bring me a bag of snack |
| Bring me a bag of chips and something to wipe a spill |
| Bring me a bag of chips and something to drink |
| Bring me a bag of chips and a soda |
| Human: I want a soda that is not coke, and a fruit. |
| I want a fruit and a soda |

Table 9: **List of ambiguous SayCan instructions**

#### A.4.2 Language Model

For planning, we use PaLM [11], a 540B parameter language model trained on a large datasets that include high-quality web documents, books, Wikipedia, conversations, and GitHub code. Before planning, we use InstructGPT [54] (`text-davinci-002`), accessed through OpenAI API. to generate the (grounded) chain of thought.

We used square bracket to indicate grounded decoding, as illustrated in Fig. 5. The prompts are shown in Listing 3.

#### A.4.3 Low-level Primitives

We use a combination of learned and scripted control policies for navigation and manipulation, following the implementation described in SayCan [2] and RT-1 [7]. The manipulation policies for the picking action are learned using Behavior Cloning (BC) on 68000 demonstrations and 12000 autonomous successes that were collected over the course of 11 months using a fleet of 10 robots. The demonstrations are collected by teleoperators using VR headset controllers to track the motion of their hand, which is then mapped onto the robot's end-effector pose. The navigation policies are scripted, based on a ground-truth map as well as a learned perception module for collision avoidance and planning. The placing actions follow pre-computed motions only when preceded by a navigation policy. The Value Functions used by SayCan for affordance grounding are provided by the $Q$-networks of trained RL agents; we follow the RL training setup described in [2].

#### A.4.4 Open-Vocabulary Detector Grounding Function

We use owl-vit [50] as our grounding model. It takes in an image and a natural language query, and returns a list of bounding boxes with scores. We take the maximum score a the grounding function.

More examples of object detection as a grounding function can be found in Fig. 9.

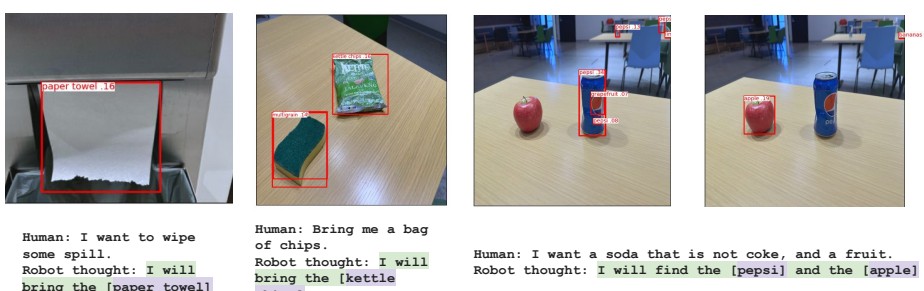

Human: I want to wipe some spill.
Robot thought: I will bring the [paper towel]

Human: Bring me a bag of chips.
Robot thought: I will bring the [kettle chips]

Human: I want a soda that is not coke, and a fruit.
Robot thought: I will find the [pepsi] and the [apple]

Figure 9: Additional examples of using open-vocabulary object detection as a grounding function in Real-World Kitchen Mobile Manipulation Domain.

## A.5 Prompts

Listing 1: Grounded Decoding Prompt in Simulated Tabletop Rearrangement Domain

```
Task: Pack all letter objects on the brown box
Step 1: pick up the e and place it on the brown box
Step 2: pick up the g and place it on the brown box
Step 3: done

Task: Put the letters on the tables in alphabetical order
Step 1: pick up the c and place it on the bottom left side
Step 2: pick up the d and place it on the right of c
Step 3: pick up the i and place it on the right of d
Step 4: pick up the l and place it on the right of i
Step 5: pick up the w and place it on the right of l
Step 6: done

Task: Spell as much of "blue" as you can
Step 1: pick up the l and place it on the bottom left side
Step 2: pick up the the u and place it on the right of l
Step 3: pick up the the e and place it on the right of u
Step 4: done

Task: Separate the vowels from the remaining letters
Step 1: pick up the i and place it on the bottom side
Step 2: pick up the o and place it on the bottom side
Step 3: done

Task: Stack all the blocks
Step 1: pick up the brown block and place it on the pink block
Step 2: pick up the cyan block and place it on the brown block
Step 3: pick up the orange block and place it on the cyan block
Step 4: pick up the gray block and place it on the orange block
Step 5: done

Task: Put all the blocks on the bottom left corner
Step 1: pick up the white block and place it on the bottom left corner
Step 2: pick up the yellow block and place it on the bottom left corner
Step 3: pick up the green block and place it on the bottom left corner
Step 4: pick up the blue block and place it on the bottom left corner
Step 5: pick up the purple block and place it on the bottom left corner
Step 6: done

Task: Put all the blocks in the bowls with matching colors
Step 1: pick up the cyan block and place it on the cyan bowl
Step 2: pick up the purple block and place it on the purple bowl
Step 3: pick up the brown block and place it on the brown bowl
Step 4: pick up the pink block and place it on the pink bowl
Step 5: done

Task: Pack the items into any box
Step 1: pick up the donut stick and place it on the red box
Step 2: pick up the pepsi and place it on the brown box
Step 3: pick up the peach and place it on the brown box
Step 4: pick up the strawberry and place it on the red box
Step 5: done

Task: Pack the items on the table into the brown box
Step 1: pick up the knife and place it on the brown box
Step 2: pick up the plum and place it on the brown box
Step 3: pick up the pepsi and place it on the brown box
Step 4: pick up the cupcake and place it on the brown box
Step 5: done

Task: Pack the items on the table into the brown box
Step 1: pick up the i and place it on the brown box
Step 2: pick up the green block and place it on the brown box
Step 3: pick up the l and place it on the brown box
Step 4: done

Task: Can you put some snacks on the right side for me?
Step 1: pick up the plum and place it on the right side
Step 2: done

Task: Can you pack my picnic box for me?
Step 1: pick up the orange and place it on the picnic box
Step 2: pick up the diet pepsi and place it on the picnic box
Step 3: pick up the knife and place it on the picnic box
Step 4: done
```

Listing 2: Grounded Decoding Prompt in MiniGrid 2D Maze Domain

```
You are a 2D maze-solving agent with access to a variety of low-level skills such as picking up or
    dropping objects, navigating to doors/keys/boxes, and opening/closing doors. Here are some example
    tasks:

Task: get the green key from the purple room, unlock the green door and go to the goal
Step 1: go to the purple door and open it
Step 2: go to the green key
Step 3: pick up the key
Step 4: go to the green door and open it
Step 5: go to the goal.

Task: pick up the purple box
Step 1: go to the green obstacle
Step 2: pick up the obstacle
Step 3: place the obstacle
Step 4: go to the blue key
Step 5: pick up the blue key
Step 6: go to the blue door and open it
Step 8: drop the blue key
Step 7: go to the purple box
Step 8: pick up the purple box.

Task: traverse the rooms to get to the goal
Step 1: go to the purple door and open it
Step 2: go to the green door and open it
Step 3: go to the purple door and open it
Step 4: go to the green door and open it
Step 5: go to the green door and open it
Step 6: go to the goal.

Now your turn.
```

Listing 3: Grounded Decoding Prompt in Real-World Kitchen Mobile Manipulation Domain

```
The following objects are in the scene: 7up, apple, banana, mango, tea, multigrain chips, kettle chips,
     jalapeno chips, rice chips, coke, grapefruit soda, pepsi, redbull, energy bar, lime soda, sponge,
     paper towel, and water bottle.
The following locations are in the scene: close counter, far counter, table, trash, bowl.
The robot will always put object name in brackets [].

Robot: I am a robot that can bring objects to you.
Human: I am hungry.
Robot thought: I will find the [multigrain chips].
Robot plan: 1. Find the multigrain chips
2. Pick up the multigrain chips
3. Bring it to you
4. Put it down
5. Done

Robot: I am a robot that can bring objects to you.
Human: Throw away the fruit.
Robot thought: I will find the [mango] and move it to the trash.
Robot plan: 1. Find the mango
2. Pick up the mango
3. Go to the trash
4. Put it down
5. Done

Robot: I am a robot that can bring objects to you.
Human: (inject instruction).
Robot thought:
```

