# OpenReview forum: "Grounded Decoding: Guiding Text Generation with Grounded Models for Embodied Agents"
_NeurIPS.cc/2023/Conference — NeurIPS 2023 poster_

### Official Review · Reviewer_TJzA · 2023-06-19

**Soundness:** 3 good
**Presentation:** 2 fair
**Contribution:** 2 fair
**Rating:** 6
**Confidence:** 3

**Summary:**

The paper introduces a novel method to tackle a key challenge in utilizing Internet-scale knowledge from LLMs (Language Model Models) in robotics tasks. Specifically, the focus is on aligning the instructions generated by LLMs with the actual actions performed to manipulate objects. The paper endeavors to frame this issue in a manner analogous to probabilistic filtering. This framing aims to decode a sequence that exhibits both high probability according to the language model and high probability according to a predefined set of objectives or attainable skills in the grounded model.

**Strengths:**

The paper introduces a novel formulation focused on decoding language models in a robot-centric manner to facilitate long-horizon robotics tasks using token-conditioned grounded models. Additionally, the paper provides empirical evidence showcasing the efficacy of this approach in comparison to CLIPort, which utilizes CLIP as an open vocabulary knowledge-based grounding technique combined with the Transporter Network. Furthermore, the paper demonstrates the transfer-ability of this idea to various domains, including 2D mazes and mobile manipulation.

**Weaknesses:**

1.The paper could benefit from improvements in certain sections of the writing. For instance, there is an awkward expression in the phrasing of the sentence on lines 116-117.
2.It would be valuable if the paper included a comparison with other techniques, such as Grounded SAM, which involves detecting objects in the scene and using them as input for LLMs to enable grounded reasoning. This alternative baseline would provide a more robust comparison for the proposed approach, as it still allows for the use of a planner to generate the task plan.
3.The author's efforts to demonstrate the effectiveness of their approach across different domains are appreciated. However, it would be advantageous if the author conducted real-world experiments that are reproducible with a broader range of robots and baselines.

**Questions:**

1. Would like to see comparison with other methods using detection to learn the labels of the object, and feeding those into the LLMs for grounded reasoning.
2. Implementation on more general robot framework for reproducible.


**Limitations:**

The authors have address the limitations adequately.

---

> ### Author Rebuttal · Authors · 2023-08-10
>
> We thank the reviewer for recognizing the novelty, experimental evaluations, and applicability of the proposed method to various domains. We also appreciate the constructive feedback. Please see below for our responses.
>
> >*”The paper could benefit from improvements in certain sections of the writing. For instance, there is an awkward expression in the phrasing of the sentence on lines 116-117.”*
> - Thank you for pointing this out. We will improve the writing for the camera-ready version and rephrase the sentence on lines 116-117.
>
> >*”It would be valuable if the paper included a comparison with other techniques… which involves detecting objects in the scene.”*
> - Though we agree that is a possible option, we note that such a detection is not equivalent to an affordance as described above. It would not capture the distance to objects for instance.
> - We will add a section discussing grounding functions versus prompting with tradeoffs and examples.
> - Importantly, the proposed formulation also complements what may be described in the prompt by guiding LLM generation with **continuous probabilities** from any grounding functions. This is thus more flexible, high fidelity, and general method to ground LLMs for embodied applications.
>
> >*”it would be advantageous if the author conducted real-world experiments that are reproducible with a broader range of robots and baselines”*
> - We agree that more extensive real-world experiments across broader range of robots would further enhance the paper, but we would also like to emphasize that the main contribution of GD is to present a general framework for guiding LLM-based planners with grounding functions, which is agnostic to the type of robots used. Therefore, our main experimental results are focused on validating the proposed method, with statistically significant evaluations in simulation as well as demonstration on physical hardware.

---

> > ### Comment · Reviewer_TJzA · 2023-08-10
> > **Thank you**
> >
> > Thank the authors for response, and clarifying my doubts. Would take all these into consideration.

---

### Official Review · Reviewer_XxHZ · 2023-07-01

**Soundness:** 3 good
**Presentation:** 4 excellent
**Contribution:** 2 fair
**Rating:** 5
**Confidence:** 4

**Summary:**

The paper introduces Guiding Text Generation with Grounded Models for Robot Control. Previous works have shown that large pre-trained language models can generate long action plans for robots based on abstract instructions. However, these models struggle to handle unstructured environments with large action spaces. The project aims to enable language models to generate control commands in a grounded and efficient manner.

The paper explains how auto-regressive language models generate text by scoring all possible tokens based on their likelihood of completing the current action step.

The approach incorporates affordance, preference, and safety functions to create grounded models. The decision-making process of selecting tokens is illustrated using the example of stacking blocks. The work also demonstrates how the method can generalize to different tasks without additional training. The flexibility of grounded decoding allows for including objective functions and constraints on the fly.

The project concludes by discussing the potential applications of grounded decoding in other embodied tasks and the possibility of using advanced search methods for better performance.

**Strengths:**

The work demonstrates its strength through plenty of experiments. By conducting numerous experiments (Embodied environment, maze, robot control in real space), the researchers provide a comprehensive evaluation of the proposed method and its performance in various scenarios. This enhances the robustness of the findings and allows for a better understanding of the method's capabilities and limitations.

The work takes into consideration the environmental conditions during planning. By incorporating affordance functions, preference functions, and safety functions, the grounded models consider the feasibility, preferences, and safety aspects of actions within the given environment. This ensures that the generated action plans are not only coherent with the high-level goals but also take into account the specific constraints and requirements of the environment, leading to more practical and context-aware robot control.

The key idea of the grounded model is well-illustrated. Through clear and concise explanations supported by visual representations and examples. By effectively communicating the core concept of the grounded model, it is easier to understand how the model works and how it addresses the challenges of generating control commands in a grounded and efficient manner. The illustrations help to visualize the decision-making process and the interaction between the language model and the affordance model, contributing to a more intuitive grasp of the proposed approach.

**Weaknesses:**

Even though the paper is titled with `Robot Control`, it lacks discussion of the robot motion planning considering the environment; i.e. how the robot motion planning changes when involving the grounded model. Especially, comparing the robot motion planning between Saycan and this work can be more persuasive.

The motivation for involving grounded information is not sufficient. It seems that the grounded model is a direct fusion of affordance map derived by CLIPort, while the whole system is equipped with template language instructions. A more straightforward approach would be using CV model to detect the object in the scene and then include the scene description in the prompt. For example: pick up the fruit in the scene, given `orange, apple, potato, ...`.

The key idea for considering the grounded map should be that the robot motion needs to think about its configuration, safety, environmental limitations, etc. However, those facts seemed to be disregarded.

**Questions:**

(1) What is the task size and the robot search space?

(2) Why the 2D Maze experiment matters since it is unrelated to robot control

**Limitations:**

The limitations are mentioned;
No societal impacts found

---

> ### Author Rebuttal · Authors · 2023-08-10
>
> We thank the reviewer for recognizing the number of experiments in diverse domains. We appreciate the constructive feedback, please see our responses below.
>
>
> >*Lack of robot control*
> - We recognize the comment that the title may not best suit the work and intend to amend the name to “Grounded Decoding: Guiding Text Generation with Grounded Models for **Embodied Agents**”, which we feel reflects the algorithm and experimental domains (including the mentioned 2D Maze environment).
>
>
> >*”lacks discussion of the robot motion planning considering the environment; i.e. how the robot motion planning changes when involving the grounded model. Especially, comparing the robot motion planning between Saycan and this work can be more persuasive”*
> - The low-level motion generation between this work and SayCan is the same. We make the general assumption that there is a low-level policy that can convert state into low-level actions, instead focusing on generating grounded action plans for a given instruction, connecting to said policy – which in both grounded decoding and SayCan is kept fixed. As such the significant difference lies in high-level planning, which is summarized in table 3 and 4.
>
>
> >*“The motivation for involving grounded information is not sufficient.” and “The key idea for considering the grounded map should be that the robot motion needs to think about its configuration, safety, environmental limitations”*
> - We agree that the consideration of the robot’s configuration, safety, environmental limitations are crucial to a grounding function, as are the abilities of the robot’s underlying policies and the state of the environment. We state a version of this on L109 and indeed our grounding functions *do* consider configuration and the environment. As a few examples of where our grounding is more than detection: for the tabletop manipulation experiments CLIPort’s affordance understands that a block that has been stacked on may be visible but cannot be picked, for the 2D maze experiments the value function captures that to open a door one must have a key and that door must be navigable, and for the mobile manipulator experiments the value functions capture that seen object far out of reach cannot be picked.
> - We recognize this was not as clear as it should be and will amend the paper to make this much clearer, by adding the above relevant examples to each grounding function definition (L191, L231, L264) as well as to the discussion in Section 3.4.
>
>
> >*”A more straightforward approach would be using CV model to detect the object in the scene and then include the scene description in the prompt.”*
> - Though we agree that is a possible option, we note that such a detection is not equivalent to an affordance as described above. It would not capture the distance to objects for instance.
> - We will add a section discussing grounding functions versus prompting with tradeoffs and examples.
> - Importantly, the proposed formulation also complements what may be described in the prompt by guiding LLM generation with **continuous probabilities** from any grounding functions. This is thus more flexible, high fidelity, and general method to ground LLMs for embodied applications.
>
>
> >*”What is the task size and the robot search space?”*
> - The tasks are generally 5-10 steps long, and the search space varies depending on the tasks. Additionally, we note that for 2D maze and tabletop manipulation domains, because the policies are free-form language-conditioned, the full search space can be considered the full language space. Below we list the number of valid skills for each domain. We will include this information in the updated paper.
> - 2D Maze: 78 skills
> - Tabletop Manipulation: 3477 skills (57 pick objects, 61 place locations)
> - Mobile Manipulation: ~60 skills
>
>
> >*”Why the 2D Maze experiment matters since it is unrelated to robot control”*
> - While the 2D Maze environment doesn’t directly deal with control, it is a nice platform to validate the main contribution of the paper that studies guiding LLM generation with grounded models. The environment provides intuitive affordance functions and low-level policies that isolate the robotic system designs that are orthogonal to our contributions, while containing a large search space. The methodologies developed in the 2D Maze environment can be directly applied to other embodied environments. In addition, as mentioned above, we will change the name of the paper to better reflect its focus on “Embodied Agents”.

---

> > ### Comment · Reviewer_XxHZ · 2023-08-20
> >
> > Thanks for the additional information provided. Now I understand the concept and motivation a little deeper and recognize your contribution.

---

### Official Review · Reviewer_WRL4 · 2023-07-05

**Soundness:** 3 good
**Presentation:** 3 good
**Contribution:** 2 fair
**Rating:** 6
**Confidence:** 4

**Summary:**

This paper studies grounding Large Language Models (LLMs) for robot control. The planning tokens are generated by the joint probability of an LLM and grounded models sequentially. The grounded models can be RL value functions, multi-modal perception models or human rules. As an extension of SayCan, this method allows open-vocabulary control and combining multiple grounded models. Experimental results in two robotic environments and a maze environment demonstrate the effectiveness of the proposed method for solving long-horizon complex tasks.

**Strengths:**

1. The paper is well-written, with relevant references and easy-to-follow narration.

2. The paper conducts experiments in various domains and tasks, showing that the proposed grounded decoding method can be widely applied in robotic tasks.

**Weaknesses:**

1. The contribution of this paper is mainly incremental to SayCan. Though it allows open-vocabulary planning, the key difference is that SayCan requires a fixed set of low-level skills while this paper requires a language-conditioned low-level policy.

2. For the proposed 3 types of grounded models: token-conditioned value functions are similar to the skill affordance in SayCan. Multi-modal foundation models and rule-based methods are more like technical tricks in applications rather than a systematic method.

**Questions:**

1. For the results in Table 3 and 4, does SayCan use the same LLM to your method?

2. According to Table 4, SayCan costs a large number of tokens since it computes scores for all the low-level skills. In its setting where a fixed set of low-level skills are provided, can we use the LLM to output open-vocabulary tokens and match the skill names using text embedding's similarity, to decrease the token cost of SayCan?

**Limitations:**

 The authors discussed the limitations in the paper. Some of these are addressed and others are left to future work.

---

> ### Author Rebuttal · Authors · 2023-08-10
>
> We thank the reviewer for the positive remarks on the comprehensiveness of our experimental results as well as the presentation quality. Please see below as we provide more clarification to the raised points.
>
> >*”The contribution of this paper is mainly incremental to SayCan”*
> - The main distinction between GD and SayCan is that GD grounds LLMs in *autoregressive generation*, which does not restrict it to output pre-defined skills only. Therefore, we believe GD presents a more general and flexible framework that is capable of incorporating a *larger* variety of grounding functions in a more *flexible* form that especially scales well with multi-task policies. As capabilities of multi-task policies continue to scale, we expect this difference to be even more pronounced.
>
> >*”For the proposed 3 types of grounded models: token-conditioned value functions are similar to the skill affordance in SayCan. Multi-modal foundation models and rule-based methods are more like technical tricks in applications rather than a systematic method.”*
> - Indeed. While our main contribution is presenting a general framework for grounding LLM-based planners in autoregressive generation, we do not find that there is a generalist grounded model that works well across different domains, which is also listed as a limitation in the paper. The main purpose of the presented grounded models is to serve as implementation examples that instantiates the proposed method in different domains depending on the task requirements.
>
> >*”For the results in Table 3 and 4, does SayCan use the same LLM to your method?”*
> - Yes.
>
> >*”In its setting where a fixed set of low-level skills are provided, can we use the LLM to output open-vocabulary tokens and match the skill names using text embedding's similarity, to decrease the token cost of SayCan?”*
> - Yes, this should be able to decrease the token cost of taking forward passes through the LLM. However, it has a few drawbacks. First, as the number of low-level skills combinatorially increases, computing the affordance for all of them may become intractable. Second, the LLM without the grounded context of the scene may output either too generic or incorrect skills. The embedding then needs to correct this, but without the full context of the query it may make mistakes. For instance in the tabletop domain, the LLM may output a skill to pick an orange object, but have to decide whether “yellow” or “blue” is closer. Whether the task is “pick up the lightest color object” or “pick up the darkest color object” changes which to correct the skills to. Instead, in GD the autoregressive grounding can handle this directly during decoding.

---

> > ### Comment · Reviewer_WRL4 · 2023-08-18
> >
> > Thank you for your detailed response! I will maintain the score.

---

### Official Review · Reviewer_VNBy · 2023-07-07

**Soundness:** 4 excellent
**Presentation:** 4 excellent
**Contribution:** 4 excellent
**Rating:** 6
**Confidence:** 4

**Summary:**

This paper proposes Grounded Decoding, a methodology to ground the high level semantic plans output by an LLM into a realizable action sequence. Specifically, the authors construct a model which decodes a sequence that is likely to be realizable given the current state of the robot and environment, but also likely according to the language model. Experimental results in both simulation and the real world demonstrate the utility of the proposed method over baselines from prior work.

**Strengths:**

- Well-written. The paper is well-written, and the figures aid in the understanding of the methodology.
- Strong experimental results. Show the utility of using both the ungrounded LLM and their Grounded Decoding relative to prior work on a range of tasks both in simulation and the real world. Outperform SayCan in terms of efficiency by a non-trivial margin. The inclusion of the qualitative results is also useful.
- Ablations. The experiments in simulation include ablations which isolate the contributions of the proposed grounded decoding module. Furthermore, the authors include ablations demonstrating the impact of the choice between greedy search and beam search, which is insightful.
- Inclusion of failure mode analysis. The analysis on the failure modes of the system is insightful.

**Weaknesses:**

- Limited experimental evaluation. The authors include results on just two environments in simulation and one in the real world. Showing results on a larger number of environments would make the claims more convincing.
- Lack of comparisons between grounding functions. Section 3.4 proposes three different techniques for obtaining grounded models that can be leveraged in GD’s formulation. However, the three techniques do not seem to be directly compared in the experiments. An experiment of this form may provide insight about which formulation works best for the proposed algorithm. Furthermore, how do the different grounding functions (AF, S, P) impact performance in the experiments?
- Ground truth map for navigation. The navigation module used in the mobile manipulation in a physical kitchen experiment is scripted and uses the ground truth map of the environment for navigation. This is a major assumption and limits deployability to only seen environments.
- Prompt engineering. It is claimed that a certain amount of prompt engineering was done to steer the LLMs. This is an undesirable property of the proposed system. How sensitive is the system to such prompt engineering? Was the same effort also applied in prompt engineering for the baseline methods?

**Questions:**

- In Figure 3 (leftmost), why is the grounded model's score for _orange higher than that of _red, when the red block is visible but there is no orange block present? Similarly, why is the language model's score for _yellow so high in the rightmost plot?

**Limitations:**

The authors include a detailed section on the limitations of the work.

---

> ### Author Rebuttal · Authors · 2023-08-10
>
> We thank the reviewer for the positive remarks on the comprehensiveness of our experimental results as well as the presentation quality. Please see below as we provide more clarification to the raised points.
>
> >*”The authors include results on just two environments in simulation and one in the real world. Showing results on a larger number of environments would make the claims more convincing.”*
> - We hoped over these three domains to show diversity of environments for which GD is applicable. We would attempt to include more results on our environments if there is a specific concern for a aspect of the proposed method, and would be open to new domains if there are some the reviewer can propose that have an available low-level policy.
>
> >*”Lack of comparisons between grounding functions… which formulation works best for the proposed algorithm… how do the different grounding functions (AF, S, P) impact performance in the experiments?”*
> - The main contribution of GD is presenting the general framework of guiding LLM-based task planning with grounding functions. To this end, we show different implementations in different domains leveraging different grounding functions. However, it is difficult to directly compare them as they serve different purposes depending on the tasks in each domain.
> - Additionally, to the best of our knowledge, there doesn’t exist a single grounding function that works the best across all domains. This is also listed in the limitations section of the paper.
> - “AF, S, P” refers to affordance, safety, and preference respectively. Different tasks are designed to evaluate each of these grounding functions, where this is used as part of the success criterion. Please see the complete results in Table 6 in Appendix.
>
> >*”Ground truth map for navigation… This is a major assumption and limits deployability to only seen environments.”*
> - Thank you for pointing this out. We recognize this as a limitation for our mobile manipulation implementation. However, since navigation policies are not part of our main contributions, we believe implementation of a more advanced navigation module is orthogonal to our work. We note that GD may also be extended to learn/obtain better navigation policies by generating step-by-step plans to visit likely regions in an unseen environments, where likelihood can be guided via a grounding function.
>
> >*”How sensitive is the system to such prompt engineering? Was the same effort also applied in prompt engineering for the baseline methods?”*
> - We find that more advanced language models are less sensitive to prompt engineering. As language model capabilities continue to improve, we expect the method to be less dependent on the prompt engineering effort.
> - GD works by guiding the **continuous probabilities** in the decoding process. Therefore, same prompts are used for all baseline methods. This is a major advantage of GD: under the same prompt engineering effort, incorporating grounding functions in the decoding process should likely only help generate better task plans.
>
> >*”In Figure 3 (leftmost), why is the grounded model's score for _orange higher than that of _red, when the red block is visible but there is no orange block present?”*
> - This is likely due to the imperfectness of the grounded model, as they are only trained on limited data in the environment and may confuse the “orange” with “yellow” or other colors.
>
> >*”Similarly, why is the language model's score for _yellow so high in the rightmost plot?”*
> - The language model score is very high for this token because the task is “stack all the blocks” and the previous step already stacks the yellow block onto the purple block. Therefore, the next block must be stacked on top of the yellow block in order to complete the full instruction.
> - This highlights the particular strength of GD in that it considers both semantic likelihood from LLMs from their strong reasoning capabilities as well as grounded information from grounding functions.

---

> > ### Comment · Reviewer_VNBy · 2023-08-21
> >
> > Thanks for the response. The authors' clarifications have addressed my questions.

---

### Official Review · Reviewer_pMcP · 2023-07-08

**Soundness:** 3 good
**Presentation:** 4 excellent
**Contribution:** 2 fair
**Rating:** 6
**Confidence:** 4

**Summary:**

This paper proposes Grounded Decoding (GD), a method that filters the decoded task descriptions by the probability of a grounded model, LLM-based task planning. This paper shows the framework can integrate different types of grounded models. Experiments across physical and simulated environments show that this decoding method outperforms prior models in instruction following tasks.

**Strengths:**

- The introduction of grounding functions covers a family of methods that evaluates the likelihood of the generated text plans conditioned on the current states.
- This paper presents the method clearly and it is easy to follow.
- The inclusion of grounding in the decoding process reduces the planning failure.

-----

Post-rebuttal update

The authors' responses answered my questions. I agree that GD is a more flexible framework and can be more easily extended to include various grounding functions as opposed to SayCan. I would encourage the authors to include the rebuttal responses to the paper to improve clarity.

**Weaknesses:**

- While it is possible to include safety and preference in the grounding function, the current form to include safety and preference is quite preliminary. It only considers the presence of objects but usually, these should be represented as statements.
- It is a good modification of SayCan to constrain the grounding in the brackets. It is unclear how reliable the ungrounded decoding and activate or revert grounding at the right time and how this modification fits the formulation of GD (Eq. 7).
- The grounding function can be used as the affordance function in SayCan. If we consider the grounding function this way, the main advantage of GD is its efficiency not the introduction of the grounding function.

**Questions:**

- The results show the overall success rate. How does it perform given the safety and preference grounding?
- There is a huge gap between planning and execution in the real robot experiment. What causes this big gap?
- In Fig 5, why doesn’t include SayCan’s failure breakdown?
- Fig 6 is quite confusing, the description says each dot is an instruction and they are colored by the affordance value computed against four different scenes. How are those scenes selected and why some instructions are infeasible if they are meaningful instructions for that domain?

**Limitations:**

The appear discussed the limitation of the proposed method well. The joint decoding may be limited as the grounding models are limited and most of them are object-centric grounding.

---

> ### Author Rebuttal · Authors · 2023-08-10
>
> We thank the reviewer for the constructive feedback. Please see our responses below.
>
> >*”the current form to include safety and preference is quite preliminary… these should be represented as statements”*
> - The main contribution of GD is presenting the general framework of guiding LLM-based planning with grounding functions, for which we extensively study affordance grounding. At the same time, we seek to show the generality of the concept and extend to additional use cases with safety and preference guidance, where we adopt the simplest methods for implementing these functions. We believe a more sophisticated implementation is very complementary to our work, and we leave this and its in depth study for future work.
> - We’d like to highlight though, that the framework of GD goes beyond putting statements in the prompt, because **continuous probabilities** are used to ground LLMs. This means that more complex safety or preference can be incorporated that leverage other modalities and are more expressive than prompting. For example, 1) continuous probabilities generated by safety or preference models from *other modalities* (e.g., images input into a CLIPort or PPO policy used in affordance grounding), or 2) continuous probabilities generated by another LLM by reading long documents of safety manuals or user history data.
> - We will add a section discussing grounding functions versus prompting with tradeoffs and examples to the updated paper.
>
> >*”It is unclear how reliable the ungrounded decoding and activate or revert grounding at the right time and how this modification fits the formulation of GD”*
> - We find the behaviors to be quite reliable given appropriate prompting – this was not a failure mode. We note that for less proficient language models, it may be less consistent. We will add a statement to this effect.
> - This modification fits into the formulation by setting the grounding function outside of the brackets to identity, as all statements are possible (and thus “grounded”) outside of the objects in the scene. We will clarify this in the paper.
>
> >*”The grounding function can be used as the affordance function in SayCan… the main advantage of GD is its efficiency not the introduction of the grounding function”*
> - The main distinction between GD and SayCan is that GD grounds LLMs in **autoregressive generation**, where SayCan grounds LLMs by **enumeration and ranking**, which requires a pre-defined set of skills and is restricted to output those skills only. Therefore, we believe GD presents a more general and flexible framework that is capable of incorporating a **larger** variety of grounding functions in a more **flexible** form, which especially scales well with multi-task policies. Though we agree that efficiency is a contribution of GD, as seen in Section 5.1, where GD is two orders of magnitude more efficient than SayCan. As capabilities of multi-task policies continue to scale, we expect this difference to be even more pronounced.
>
> >*”The results show the overall success rate. How does it perform given the safety and preference grounding?”*
> - The complete results of the safety and preference grounded tasks can be found in Table 6 in Appendix. For these tasks, the agent not only has to complete the task but also needs to follow the requirements to be considered “successful”. For example, GD attains a success rate of “100%” and “75%” on two tasks that require safety grounding, while the ungrounded baseline has only “5%” success rate. On one task that requires preference grounding, GD attains “55%” and ungrounded baseline has only “15%”.
>
> >*”There is a huge gap between planning and execution in the real robot experiment. What causes this big gap?”*
> - The gap comes from policy failure, where the plan generated by GD is correct but the underlying policy does not successfully complete the sub-task.
>
> >*”In Fig 5, why doesn’t include SayCan’s failure breakdown”*
> - We only show failure breakdown in the simulated tabletop domain, where it’s easier to quantitatively and statistically compare to different baselines over a large number of experiments. As real SayCan evals are quite time consuming to run, we feel we do not have enough data points to generate such a figure.
>
> >*”In Figure 6… How are those scenes selected and why some instructions are infeasible if they are meaningful instructions for that domain?”*
> - Thank you for pointing this out. Please see below for clarification, and we will also improve the clarity in the paper.
> - The scenes are manually selected to show as illustrative examples on why GD is efficient at identifying and grounding actions. More specifically, the domain refers to the general tabletop domain, where different initializations with different sets of objects are possible. We consider an instruction “meaningful” as long as they follow the pick&place action template and refer to possible objects in the full domain. However, depending on the specific initializations, most objects will not be present for each scene, thus most instructions are infeasible (even though they might be “meaningful” in some other scenes in the same domain).

---

### Decision · Program_Chairs · 2023-09-21

**Decision:**

Accept (poster)

**Comment:**

This paper proposes Grounded Decoding (GD), a method to ground language model text generation for robot control by incorporating additional "grounding" information. GD decodes sequences that are likely under both a language model and grounded models like affordances or safety functions.

Several reviewers recommended acceptance, citing the novelty of the grounding formulation, strong experimental results across domains, efficiency over prior methods, and intuitive explanations. There were a few concerns mainly about the possibly incremental nature of improvement over prior work like SayCan. Reviewer XXHZ gave a borderline accept, while also commenting that 'robot control' might be a bit of a stretch in describing the paper. The authors acknowledged this comment with their proposal to rename it to 'embodied control', expanding the scope towards more conceptual environments such as maze2D. The reviewer-author discussion also highlighted the increased flexibility over SayCan, and the focus on high-level planning.

Most of the other issues raised such as limited environments being tested, reliance on prompt engineering, etc. were addressed by the authors by highlighting flexibility of grounding functions, reduced prompt dependence of advanced LLMs, etc. I believe the proposed scheme is a novel enough formulation and the evaluations reasonably validate the approach, even if additional environments could strengthen claims. I recommend acceptance and encourage the authors sufficiently clarify novelty and scope in the camera ready (for example, adding the rebuttal discussion points in the paper). Expanding analysis of grounding choices would also be beneficial.